# Manufacturing of Aluminum Nano-Composites Reinforced with Nano-Copper and High Graphene Ratios Using Hot Pressing Technique

**DOI:** 10.3390/ma16227174

**Published:** 2023-11-15

**Authors:** Hossam M. Yehia, Reham A. H. Elmetwally, Abdelhalim M. Elhabak, Omayma A. El-Kady, Ahmed Yehia Shash

**Affiliations:** 1Mechanical Department, Faculty of Technology and Education, Helwan University, Cairo 11795, Egypt; 2Faculty of Engineering, Cairo University, Cairo 12613, Egypt; 3Powder Technology Department, Central Metallurgical Research and Development Institute (CMRDI), Helwan 11421, Egypt; o.alkady68@gmail.com; 4Faculty of Engineering and Materials Science, German University in Cairo, New Cairo 11511, Egypt

**Keywords:** nano-aluminum, nano-copper, electroless coating, graphene, hardness, compressive strength

## Abstract

In this study, the nano-aluminum powder was reinforced with a hybrid of copper and graphene nanoplatelets (GNPs). The ratios of GNPs were 0 wt%, 0.4 wt%, 0.6 wt%, 1.2 wt% and 1.8 wt%. To avoid the reaction between aluminum and graphene and, consequently, the formation of aluminum carbide, the GNP was first metalized with 5 wt% Ag and then coated with the predetermined 15 wt% Cu by the electroless coating process. In addition, the coating process was performed to improve the poor wettability between metal and ceramic. The Al/(GNPs-Ag)Cu nanocomposites with a high relative density of 99.9% were successfully prepared by the powder hot-pressing techniques. The effects of (GNPs/Ag) and Cu on the microstructure, density, hardness, and compressive strength of the Al-Cu nanocomposite were studied. As a result of agitating the GNPs during the cleaning and silver and Cu-plating, a homogeneous distribution was achieved. Some layers formed nano-tubes. The Al_4_C_3_ phase was not detected due to coating GNPs with Cu. The Cu_9_Al_4_ intermetallic was formed during the sintering process. The homogeneous dispersion of Cu and different ratios of GNs, good adhesion, and the formation of the new Cu_9_Al_4_ intermetallic improved in hardness. The pure aluminum sample recorded 216.2 HV, whereas Al/Cu reinforced with 1.8 GNs recorded 328.42 HV with a 51.9% increment. The compressive stress of graphene samples was improved upon increasing the GNPs contents. The Al-Cu/1.8 GNs sample recorded 266.99 MPa.

## 1. Introduction

Composite materials are classified depending on the type of matrix material into Metal Matrix Composites (MMC), Polymer Matrix Composites (PMC), and Ceramic Matrix Composites (CMC). In the metal matrix composite, the matrix is pliable and formidable, while reinforcing materials are rigid and robust. The ideal manufacturing composites combine the stiffness of the matrix and the reinforcement strength that have specific properties which do not belong to either alone. For example, aluminum metal and its alloys have low density, high elasticity, and deformability but high thermal expansion coefficients, low abrasion resistance, and relatively low strength. These characteristics reduce their applications in many fields requiring good mechanical and thermal properties. Aluminum nanoparticles are intensely specialized in various areas, including pyrotechnic, propellant, and explosive industries. Aluminum powder is added to multiple compositions to enhance their performance by raising reaction energies, flame temperatures, and increasing blast rates. Aluminum nanoparticles are more preferred due to their high combustion enthalpy and rapid kinetics, which enhance these reaction properties. Nano-sized aluminum particles represent a new energetic material with an increment in reactivity because of its large surface area. They are expected to be applied to a next-generation propellant in aerospace applications. The high reactivity of aluminum nanoparticles is based on the particle diameter. The most sensitive diameter is 30–50 nm [1]. It is lightweight for being just one-third the steel density. Like copper, it has high heat and electric conductivity, excellent corrosion resistance in most conditions, and can be easily cast or made into a wide range of consumer goods. The mechanical, physical, and chemical properties of aluminum (Al) and its nanoparticles are necessary to practice various applications, including alloy powder metallurgy parts for automobiles and aircrafts. In addition, heat shielding coatings for aircraft, corrosion, resistant, conductive, and heat reflecting paints, conductive and decorative plastics, soldering and termite welding, pyrotechnics, and military applications (rocket fuel, igniter, smokes, and tracers). Nanoscale Al particles are also studied as high-capacity hydrogen storage materials [2,3]. Copper metal (99.9%) also has high and efficient electric conductivity, but at the same time, it has poor mechanical characteristics such as hardness and compression. The copper nano-powder has particle sizes of less than 100 nm. Copper nano-powder is widely used for electronics, catalysts, and pigments. The potential needs for nanomaterials show an increasing trend [4]. In addition, copper (Cu) nanoparticles were used as alternatives for other noble metals for many applications, such as heat transfer and inkjet printing. Inkjet printing is the most promising technology which uses noble metals such as gold and silver to print highly conductive elements. Compared to copper, the cost of noble metals is very high. Therefore, due to its low cost and high conductivity, copper is highly preferred for producing highly conductive Cu patterns on a plastic substrate through inkjet printing [5,6].

Adding graphene, CNT, SiC, WC, or graphite in powder form to aluminum or copper metals improves their mechanical characteristics without affecting their physical properties. From an engineering point of view, graphene and its derivatives have been a focus of studies due to the novel properties of graphene and its potential use in multiple application fields. That is not limited to electronics, heat transfer, biosensing, membrane technology, battery technology, and advanced composites. Graphene has interesting features, such as large electric and thermal conductivity, high mechanical strength, and high optical transparency. These properties are only detected for graphene films formed of only one or a few layers [7,8,9]. According to Changgu Lee, graphene is the strongest material ever tested, with a tensile strength of 130 GPa and a Young’s modulus (defines stiffness) of 1 TPa [10]. Apart from this, graphene is unbelievably light, weighing only 0.77 mg/m^2^ [11]. It has been experimentally proven that its electron mobility is almost independent of temperature [12]. Graphene produces a highly opaque atomic monolayer in a vacuum, as it can absorb approximately 2.3% of white light. Adding another layer of graphene increases the amount of white light absorbed by approx. the same value (2.3%) [13].

A. Fathy et al. [14] studied aluminum reinforced with Al_2_O_3_-coated nickel and graphene nanoplatelets (GNPs) using the electroless deposition and hot-pressing technique. The compressive strength of the composite containing 1.4% GNPs was 1.52 times greater than that of pure Al. Increasing the GNPs content reduced the wear rate of Al-Al_2_O_3_/GNPs nanocomposite to 19 times less for a sample containing 1.4% GNPs than for Al. A. Saboori et al. [15] studied the densification of Al matrix reinforced with various GNPs content (0, 0.5, 1.0, 1.5 and 2.0 wt%). The samples were sintered at different temperatures (540, 580 and 620 °C) under nitrogen flow to determine the sinterability of the nanocomposites. It was determined that by increasing the temperature, the sinterability of the nanocomposite increases due to easier diffusion. On the contrary, the sintering ability of the nanocomposite decreased with the increase in GNPs content up to 2 wt% due to the work-hardening effect caused by the GNPs. The Vickers hardness of sintered nanocomposites increased proportionally with GNPs content and sintering temperature. Duosheng Li et al. [16] successfully fabricated GNPs/Al nanocomposites containing 0.5 vol%, 1.0 vol% and 2.0 vol% of GNPs. It was determined that the lower content of GNPs in the composite induced better mechanical properties and better microstructural characteristics than pure aluminum. Adding 0.5 vol%, 1.0 vol% and 2.0 vol% GNPs to the Al matrix provided the average compressive strengths of GNPs/Al nanocomposites of 297, 345 and 527 MPa, respectively, which remarkably increased the strength over the original aluminum by 330–586%. Gang Li et al. [17] studied the graphene nanosheet (GN)-reinforced Al matrix composites fabricated by high-energy ball milling and vacuum hot-pressing technique. The graphene content was 0.25 wt%, 0.5 wt% and 1.0 wt%. A good interfacial bonding was obtained in GNs/Al composites. A total of 38.27% and 56.19% increments in the yield strength and ultimate tensile strength of Al-0.25GNSs compared with pure Al were achieved. The elongation slightly decreased.

Furthermore, one of the common problems that limit the graphene application in Al (MMCs) is the poor wettability between them, which causes a low interface adhesion resulting in poor mechanical and thermal properties [18,19,20]. At the same time, the interaction between Al and C to form Al_3_C_4_ intermetallic phase reduces the mechanical properties [19,21]. The electroless deposition method is one of the solutions that can be used to produce a wettability layer on the metal’s surface to improve the interfacial bond between the metal and ceramic phases [22,23,24,25].

The density, hardness, tribological properties, and electrical conductivity of copper matrix composites reinforced with coated and uncoated Al_2_O_3_ were studied. The samples were fabricated using the hot-pressing process. In spite of decreasing the density of samples, the hardness was increased due to the hard structure of Al_2_O_3_ particles. The highest hardness was achieved for silver-coated copper matrix composites fabricated by using 2 h of milling at 3 wt% Al_2_O_3_ ratio. The highest electrical conductivity (88% IACS) was measured for the 3 wt% Al_2_O_3_ silver-coated copper sample synthesized by 2 h milling. The wear mechanism was abrasive wear with grooves and scratches [26,27,28].

Hot pressing is the setup in uniaxial hot pressing that is very similar to conventional PM pressing, except that heat is applied during compaction. The product is generally dense, intense, challenging, and dimensionally accurate [29,30,31,32,33,34,35,36]. Despite these advantages, the process presents some technical issues that limit its adoption. Principal among these is (1) selecting a suitable mold material that can withstand the high sintering temperatures; (2) a longer production cycle required to accomplish sintering; and (3) heating and maintaining atmospheric control in the process.

Because of the aforementioned knowledge, this work deals with the preparation of the aluminum–copper–graphene nanocomposites using electroless deposition and hot-pressing techniques. Scanning electron microscope (SEM) and X-ray diffraction (XRD) were used to study the formation of any new phases and the morphology of nanoparticles of the as-prepared composites. Density, hardness, and compressive strength were also estimated.

## 2. Experimental Work

### Materials and Method

The powder metallurgy–hot pressing technique was used to prepare nanocomposites consisting of graphene nano-platelets (GNPs) as the strengthening material of the Al-Cu hybrid matrix. The raw powder material specifications are as follows:The Al with ~100 nm and 99.99% purity;The nano-Cu deposited by the electroless plating process;The graphene nano-platelets (GNPs) with thickness of less than 10 nm and purity of 99.99%.

The Al was supplied from Loba Chemie Pvt. Ltd., Mumbai, India and the GNPs powder was supplied from advanced chemical supply (ACS) materials. The nano-Cu powder was prepared using copper sulfate, potassium sodium tartrate, sodium hydroxide, and 28% formaldehyde.

The particle shape and size of the raw materials were characterized by the Scanning Electron Microscope (SEM), as shown in Figure 1. The micrograph shows that the Al powder has a spherical shape and nano-size according to the measurement scale. The graphene has a sheet shape, and its thickness is in the size of the nanometer. The edges of the uncoated graphene look very sharp. Image d reveals that the GNPs layers become unbound after the cleaning and Ag-coating process, and their edges disappear after the coating process. The electroless coating process established homogeneous distribution of the Ag. Some single GNPs sheets were rolled in the form of carbon nano-tubes during the coating process. This may be happened due to the large surface area of the graphene layer compared with its thickness. Furthermore, coating with copper increases the separation of GNPs layers. Before adding the reducing agent to the copper solution, the graphene layers were stirred for 20 min, which encourages its exfoliation. It appears transparent in the image (d). Deposited graphene layers with copper metal are evident from image (e). Copper is included in the form of a dendritic shape.

One of the critical problems limiting the application of graphene in Al MMNCs is the poor wettability between the graphene and Al, which leads to a low interface adhesion, and thus poor mechanical and thermal properties. In addition, the interaction between Al and C, which forms the Al_3_C_4_ intermetallic phase, reduces mechanical properties. The electroless deposition method has been proposed to produce a wettability layer on the metal surface to improve the interfacial bond between metal and ceramic phases. Due to its low cost and suitability for any profile, the electroless deposition method is widely used in surface modification of various ceramic materials, such as non-conductors and semiconductors. This technique has also attracted a lot of interest in nano-fabrications in optics and the decoration of carbon nanotubes, SiC, and other kinds of powder. The silver electroless deposition has been performed in two stages.

The production process of the new materials began with cleaning and Ag mineralization of GNPs layers, and then precipitation of Cu metal on their surfaces with the electroless process. In the final mixing, the predetermined values of the composite’s constituents Al, hybrid (Cu-GNPs), were used. Figure 2 illustrates the production steps.

The first stage was sensitization. The GNPs were sensitized to remove any contamination that might prevent the Ag precipitation on GNPs surfaces. This process was established by submerging GNPs in 10% sodium hydroxide solution and stirring for 1 h, then immersing in acetone for 1 h. Afterward, the GNPs were filtered and washed with distilled water and dried at 100 °C for 2 h in an electric furnace.

The second stage was Ag metallization. The electroless plating process was used to coat the cleaned GNPs with 5 wt% of nano-Ag. The GNPs was stirred for 1 h in a chemical solution bath composed of 3 g/L of silver nitrate, 300 mL/l formaldehyde, and 33% ammonia solution for adjusting the PH to 12. After adding formaldehyde, the interaction started. GNPs were filtered, washed using acetone, and dried in a furnace at 100 °C for 2 h [23,24].

The third step was the precipitation of Cu nano-size particles on (GNPs/Ag) using the electroless deposition technique. The electroless plating baths are based on formaldehyde-based reducing agent and its derivatives and complexing agent, which plays an essential role in non-electroplating copper baths to obtain good quality deposits.

Complexing agents minimize the formation of Cu to (Cu(OH)_2_) in an alkaline pH range 17). They also stabilize the baths and increase bath life. The addition of complexing agents in a small quantity increases the plating rate. The chemical composition bath started with 170 g/L potassium sodium tartrate, increasing it to 35 g/L CuSO_4_. The sodium hydroxide was added with ~50 g/L to adjust the pH = 12.5. The (GNPs/Ag) with 0.4 wt%, 0.6 wt%, 1.2 wt% and 1.8 wt% were stirred for 20 min before adding the formaldehyde with 200 mL/L as a reducing agent for copper sulphate solution in an alkaline tartrate path. Figure 3 shows the sequence of the electroless deposition. It was determined that after adding the potassium sodium tartrate, the distilled watercolor became white. After adding CuSO_4_, the color of the water converted to light blue. The heat increased to 60 °C. Once the GNPs were added and stirred, the color converted to dark blue. Finally, the Cu was precipitated on GNPs, and the blue color of the bath changed gradually to colorless by adding the formaldehyde.

The precipitated Cu and GNPs weights for each composite were determined according to the mixture rule, as shown in Table 1. The (CuO/GNPs) powder was filtered and dried at 70 °C for 3 h. The CuO was reduced to Cu metal by heating at 450 °C for 120 min in an electrical furnace under a hydrogen control atmosphere.

The fourth step was mixing of Al powder with 0, 0.4, 0.6, 1.2 and 1.8 wt% GNPs/Ag coated 15 wt% nano-Cu. The mixing process was achieved using a high-energy ball milling technique using alumina ceramic balls of 12 mm diameter in an acetone medium. The milling speed was 250 rpm. The ball-to-powder ratio was 12:1 and set to be constant to promote the cold welding of the nano-Al on the GNPs coated with Ag and Cu. The milling time was 6 h to ensure homogenous powder slurry.

The last step was the hot pressing for the obtained mixed nanocomposite powders using a uniaxial hydraulic press. The process was performed by putting the powder in the W320 die steel with a 10 mm inner diameter and heating it to 550 °C for 1 h after that hot pressing at 800 MPa. Table 2 shows the chemical composition of the W320 steel.

## 3. Composite Characterization

The theoretical and actual densities of the composite powders and hot compacted samples were determined by applying the rule of the mixture and Archimedes’ rule, respectively, according to Equations (1) and (2).
(1)$$\rho_{th.}=\rho_{1}{wt\%}_{1}+\rho_{2}{wt\%}_{2}+\rho_{3}{wt\%}_{3}+\ldots ,$$
(2)$$\rho_{act.}=\frac{Wa}{Wa-Ww}\rho_{Liquid},\;\;\;\mathrm{gm}/{\mathrm{cm}}^{3}$$
where $\rho_{act.}$ is the actual density (gm/cm^3^), *W_a_* is the mass of the sample in air and *W_w_* is the mass of the sample in water (gm). (ρ*_th_*) is the theoretical densities for the exhibited composites.

The hot compacted samples were initially ground with recommended 220, 400, 600, 800, 1000, 1200 and 2000 grit SiC papers. Then, the specimens were polished using alumina paste on machine model “Buhlertm”.

The polished samples’ microstructure was investigated using the Scanning Electron Microscope (SEM) model “FE-SEM: QUANTAFEG250, Holland”. The Quanta SEM systems were supported with energy dispersive X-ray (EDAX) spectrometers, secondary electron (SE), and backscatter electron (BSE) detectors for the determination of particle size, morphology, and chemical composition of samples. The X-ray diffraction analysis was used to investigate the elemental composition and phase structure of the hot compacted samples using the X-ray diffractometer model x, Per PRO PANalytical with Cu Kα radiation (λ = 0.15406 nm).

The prepared specimens’ micro-hardness was obtained using a Vickers micro-hardness tester (Leco LM70-USA) under a test load of 500 g for 10 s according to ASTM standard E 92. The average of five indentations applied for each specimen at different places was used to express the hardness value. The compression strength of the prepared specimens was estimated at room temperature using a micro-computer-controlled uniaxial universal testing machine (HT-9501 computer servo-hydraulic universal testing machine). The samples used for the compression test were circular, with *a* 10 mm diameter and 9 mm height. The applied speed of the cross-head universal test machine was 0.9 mm/min.

## 4. Results and Discussion

### 4.1. XRD Observation

The XRD patterns of Al, Al-15 wt% Cu, Al-15 wt% Cu/xGNPs (x = 0, 0.4%, 0.6%, 1.2% and 1.8 wt%) are shown in Figure 4. The elemental analysis has been established in a 2θ range from 20 to 100°. From the first fabricated sample, the peaks of the aluminum matrix were detected at 2θ equal to ~38.56, ~44.8, ~65.28, ~78.24 and ~99.5°. Because the aluminum powder was in the nano-size, its peak intensity was not sharp. The aluminum oxide compound was detected in the pure aluminum sample. The formation of aluminum oxide with pure aluminum may be due to the heating of aluminum powder inside the W320 steel die at 550 °C, which led to the interaction of aluminum with oxygen and the formation of aluminum oxide. This also may be due to executing the process in an uncontrolled atmosphere. In the second sample, the Cu peaks appeared at 2θ equal to ~43.25, ~50.56 and ~74.24 with lower intensity than aluminum due to its low percentage and the interaction with aluminum. A new phase was observed at 2θ equal to 44.32, corresponding to Cu9Al4 intermetallic formed during the sintering process by 0.83 eV formation energy according to the phase diagram between Cu and Al [33].

The addition of graphene hindered the interaction between Al and Cu. The Cu peak at ~43.25 became visible starting from adding graphene at 0.4 wt% to 1.2 wt%. The good spread of graphene flakes can explain this due to its good agitation during various coating processes, which contributed to its good distribution and hindered the interaction between Al and Cu. The Cu peaks began to decrease again at 1.8 GNP, confirming the interaction between Al and Cu. This could be because the graphene agglomerated and was not well distributed, which facilitated the bonding between Al and Cu again. Broad peaks for all elements are observed due to their nano-grain size. The lowest intensity peaks correspond to the C because of its more moderate content. It was observed that the peaks of C became clearer and had a higher intensity in the sample containing 1.8% GNPs at 2θ equal to 26.72.

### 4.2. Density Measurements

The relative density (RD) is an essential parameter. It has excellent effects on the mechanical and physical properties of fabricated materials. Table 3 illustrates the relative density of Al, Al-Cu, and Al-15%Cu/GNPs nanocomposites as a function of GNPs content after hot compaction. The results show that the pure Al sample recorded 99.6% and increased to 99.9% for composites containing 15 wt%. The increase in density by adding Cu to Al is due to two reasons. The first is that the density of the Cu particles is greater than the density of the Al particles. Therefore, replacing higher-density particles with lower-density particles increases the density of the substance produced. The second reason, which helped to increase the density, was the good coalescence and the occurrence of a reaction between Al and Cu, as shown by the X-ray chemical analysis of the produced samples in Figure 4. Adding the GNPs to the Al-15Cu gradually decreases the density from 99.5 at 0.4 wt% to 98.7 at 1.8 wt%. The reduction in the relative density was not high regardless of increasing the GNP percent to 1.8%, which means the absence of GNP accumulation due to the long stirring time during the silver and copper precipitating using the electroless process. Appropriate mechanical mixing of GNPs with Al facilitates easy scattering of graphene atoms and diffusion between graphene nanoparticles, which results in lesser agglomeration of GNPs and higher density of fabricated nanocomposites. The hot-pressing process helped squeeze the aluminum sample, releasing any gases and removing pores between the GNP layers during manufacturing, thus improving the density. In addition, hot pressing at 800 MPa improved the densification in an excellent way that eased right consolidation via atomic diffusion along with the aluminum particle interfaces during the consolidation process. It was reported that the density decreases with increasing GNPs content [31,32]. This decrease may be due to the lower density of 2.2 g/cm^3^ of GNPs than that of Al 2.7 g/cm^3^ and Cu 8.9 g/cm^3^.

### 4.3. Microstructure Investigation

The microstructure with the BSE mode of the hot-pressed samples Al, Al-15 wt% Cu, Al-15Cu/GNPs with content (0.4, 0.6, 1.2 and 1.8 wt%) is shown in Figure 5 with low and high magnifications. Two colors from image a5 of the pure aluminum sample are observed. The first is brown for aluminum, and the second indicates the alumina phase that formed during the fabrication process, as confirmed by the x-rays. As mentioned previously, the formation of the alumina compound may have resulted from conducting the heating process of the samples in an unprotected atmosphere, which helped in the reaction of aluminum with oxygen, that is, the formation of alumina. No pores were formed due to the heat pressure. Because of the high purity of the used aluminum powder and the nano-size of the particles, no grain boundaries for aluminum particles were observed. This could be also attributed to applying the heat stress that promotes adhesion between particles and restricts particle size inflation during the production process. Image b5 illustrates the microstructure of the aluminum–copper nano-composite. Copper particles appear light-gray. An excellent distribution of Cu with Al is observed, as well as good adhesion due to the interaction between them during the hot-pressing process. The high magnification of image b5 shows that the cooper addition greatly affects the microstructure of aluminum. The microstructure consists primarily of a dendritic morphology with dendritic α-Al arms (rich in Al) and a dark phase (rich in Cu). The eutectic phase between aluminum and copper has been formed. The GNPs additions in Al-matrix affect the dendritic shape, especially at 1.2 wt% and 1.8 wt%. At 1.2 wt%, the excellent distribution of the GNPs contributes to the disappearance of the dendrite α-Al arms. Layers of the GNPs appear as a compressed nano-tube formed during the cleaning and Ag deposition process. The high magnification of Figure 5e illustrates that the GNP layers are transparent and in a horizontal position. In addition, the adhesion at the edges of the GNP layers with the Al matrix is improved due to coating of GNPs with Ag, where the wettability is improved. The addition of GNPs reveals a decrease in the respective size of dendrites. Increase in the GNPs to 1.8 wt% GNPs caused the thin layers of GNPs to be replaced the eutectic phase. Few accumulations formed as the graphene increased up to 1.8 while the tubular shape of the graphene flakes persisted.

Spatial distribution obtained by energy dispersive spectroscopy (EDS) mapping elements for principal phases of Al-15 wt% Cu/1.8 wt% GNPs are included in Figure 6. Images illustrate the presence of Al and Cu that are used as the matrix and the GNPs (C) coated with Ag. An excellent distribution for all elements is observed.

### 4.4. Hardness

Table 4 illustrates the micro-hardness of Al, Al-15 wt% Cu, and Al-15 wt% Cu/GNPs nanocomposites with the variation of GNPs content of up to 1.8 wt%. As observed, the aluminum sample recorded 216.2 HV. The pure aluminum sample’s high hardness can be attributed to two reasons. The first is the use of aluminum nanoparticles, as there is a direct relationship between the particle size and the material strength. The smaller the size of the particles, the greater the contact area between them and the less movement of dislocations, thus improving the resistive strength of the materials. The second is the formation of alumina compound during the manufacturing process, characterized by a high hardness that may reach 15 GPa [32,33,37].

The effect of particle size on the hardness of the produced samples becomes clear when we compare the hardness results of the current samples with samples in which micro-powders were used. Producing the micro-particle size at 550 °C and under 840 MPa using the hot-pressing technique reduced the hardness to 34 HV [34]. The addition of 15 wt% Cu to the pure aluminum sample raised the hardness to 230 HV. This improvement may be related to the formation of Cu9Al4 intermetallic that has exceptionally high hardness [38,39]. In addition, Cu added to Al is expected to increase the alumina content, while Al reduces the oxygenation of Cu. Increasing the temperature to 550 °C during the fabrication process leads to the oxidation of Cu that is reduced by Al, as shown in Equation (3).
2Al + 3CuO → Al_2_O_3_ +3Cu.(3)

It is evident that by increasing the GNPs content, the hardness of Al/Cu nanocomposite is gradually increased up to 1.8 wt%, in spite of decreases in the relative density. As shown in Table 3, the reduction in the relative density by varying the GNs ratios was not high, which means that the reduction may be related to only the low density of the GN layers. Porosity was absent as shown by the microstructure. The sample containing 1.8% recorded 328.24 HV compared to 216.2 HV and 230 HV for the pure Al and Al/15%Cu samples, respectively. This improvement is attributed to the GNP layer characteristics, which have extremely high strength and mechanical properties, also providing a high contact force for deforming during indentations. Another main factor that affects the hardness is the improvement of wettability between Al and GNPs by coating GNPs with Ag and Cu layers, in which proper distribution was achieved all over the Al matrix.

### 4.5. Compressive Strength

Figure 7 illustrates the room temperature compression test for all fabricated samples. A significant improvement in the compressive yield strength (CYS) and ultimate compressive strength (UCS) by adding 15 wt% Cu and different weight percentages of the GNPs to the Al matrix are observed. The results are summarized in Table 5. The Al-15%Cu/1.8 wt% GNPs nanocomposite recorded the highest CYS of 54.9932 MPa and UCS of 266.9956 MPa. All samples were fractured during the compression test. The addition of GNPs leads to work-hardening and significantly reduces fracture strain (elongation). The significant difference between the Al matrix’s intrinsic strength and the 1.8 GNPs sample may induce mismatch dislocations at the interface during compression. The high strength properties of graphene, as the tensile strength (125 GPa) and its elastic modulus (1100 GPa) are greater than the aluminum matrix. Therefore, it plays a strengthening role in the Al-15%Cu/GNPs nanocomposites. In addition, it is determined that under external loads, composite dislocations avoided the large surface area two-dimensional structure GNPs that can be considered distinguished dislocation walls which require more energy for disturbance. It is not easier to avoid the GNPs than to cut them, as they act as particulate reinforcements. The slipping of dislocations was restrained, resulting in a great effect in preventing the plastic deformation of the matrix, which showed great increases in the compressive strength of the GNPs/Al nanocomposite. The excellent adhesion, proper distribution of the GNPs in the Al matrix, unique mechanical properties of GNPs, and the high densification due to applying the hot pressing may be the main reason for improving the compression strength of the 1.8 wt% GNPs. The GNPs have wrinkled structure which effectively transfers stress from ductile Al matrix to hard GNPs, resulting in the enhanced compressive strength [15,16,17]. In addition, capsulation of GNPs by Ag and Cu layers decreases the chance of any chemical reaction with Al or Cu. Therefore, no Al_4_C_3_ is formed or any copper carbide is observed, which are undesirable phases that cause the non-wettability and create pores. This indicates that GNPs have great potential as a reinforcement material for the metal matrix composites [16,17].

## 5. Conclusions

Due to the continued stirring of graphene during the cleaning and the coating with Ag and Cu and then mixing them with the nano-aluminum particles for 6 h, a homogenous dispersion of it at high ratios was achieved.New intermetallic (Cu9Al4) between Al and Cu was formed. The formed intermetallic enhances the grain boundaries’ strength and, consequently, the fabricated aluminum matrix’s hardness.Due to encapsulating the GNs/Ag layers with Cu, the interaction between them and aluminum was restricted, and no peaks for Al_4_C_3_ were detected.The high magnification of the microstructure showed that the GNP layers were transparent and in a horizontal position. In addition, the adhesion at the edges of the GNP layers with the Al matrix was improved due to coating of GNPs with Ag, where the wettability was improved.The GNPs content significantly improved the hardness of the nanocomposite to reach 328.24 HV for samples containing 1.8% GNPs compared with 216.2 HV and 230 HV for the pure Al and Al/Cu samples, respectively.Increasing the GNPs up to 1.8 wt% GNs ratio led to increasing the compressive strength to 266.9 MPa compared with 194.43 MPa and 204.1 for pure Al and Al/15Cu nanocomposites, respectively.

## Figures and Tables

**Figure 1 materials-16-07174-f001:**
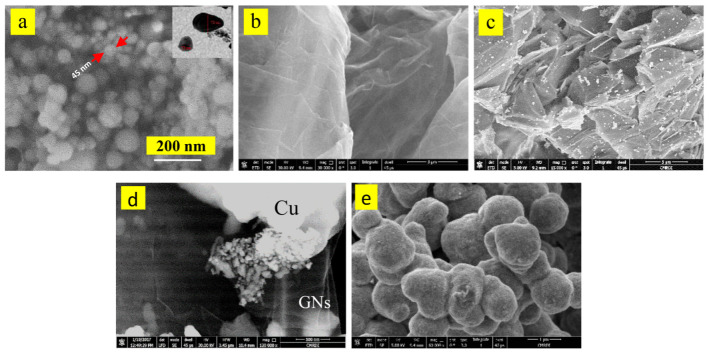
SEM (BSE) of used raw materials. (**a**) SEM of Al powder; (**b**,**c**) SEM of GNPs before and after coating with Ag; (**d**) SEM of the Cu coating GNPs; and (**e**) pure precipitated Cu.

**Figure 2 materials-16-07174-f002:**
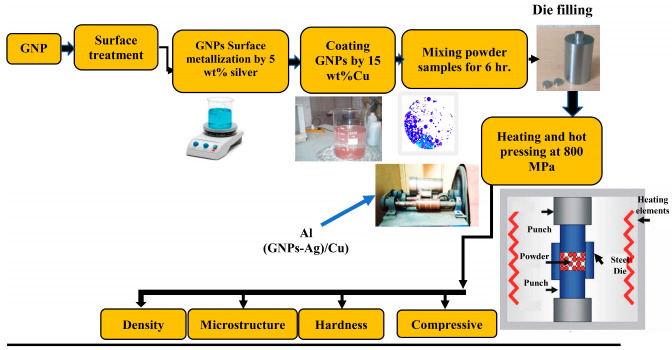
Sample preparation steps.

**Figure 3 materials-16-07174-f003:**
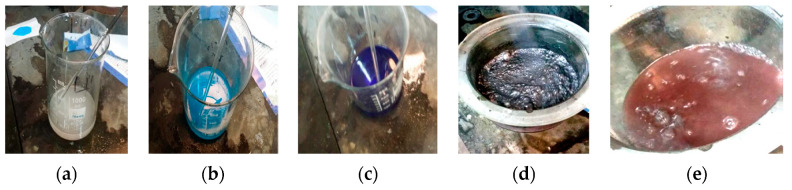
The sequence of the electroless copper deposition process (**a**) Dissolution of silver nitrate, (**b**,**c**) Dissolve copper sulfate, (**d**,**e**) Copper powder filtration.

**Figure 4 materials-16-07174-f004:**
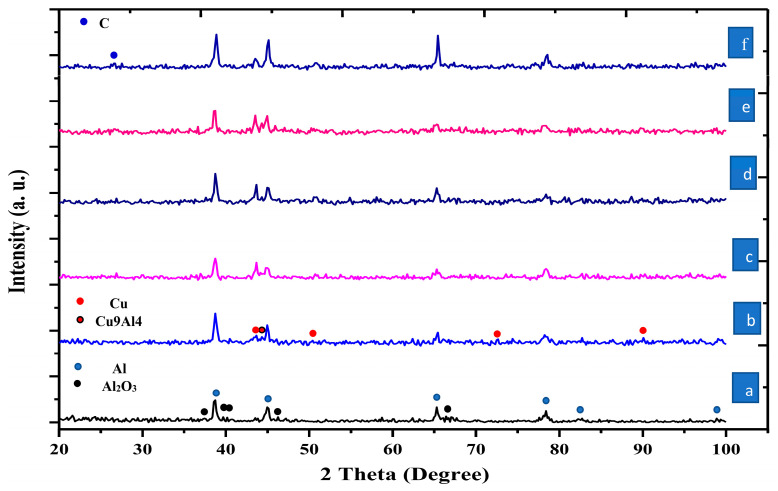
X-ray diffraction of hot-pressed materials. (a) Pure Al matrix, (b) Al-15 Cu, (c) Al-15 Cu /0.4 wt% GNPs-Ag, (d) Al-15 Cu /0.6 wt% GNPs-Ag, (e) Al-15 Cu /1.2 wt% GNPs-Ag GNPs, (f) Al-15 Cu /1.8 wt% GNPs-Ag.

**Figure 5 materials-16-07174-f005:**
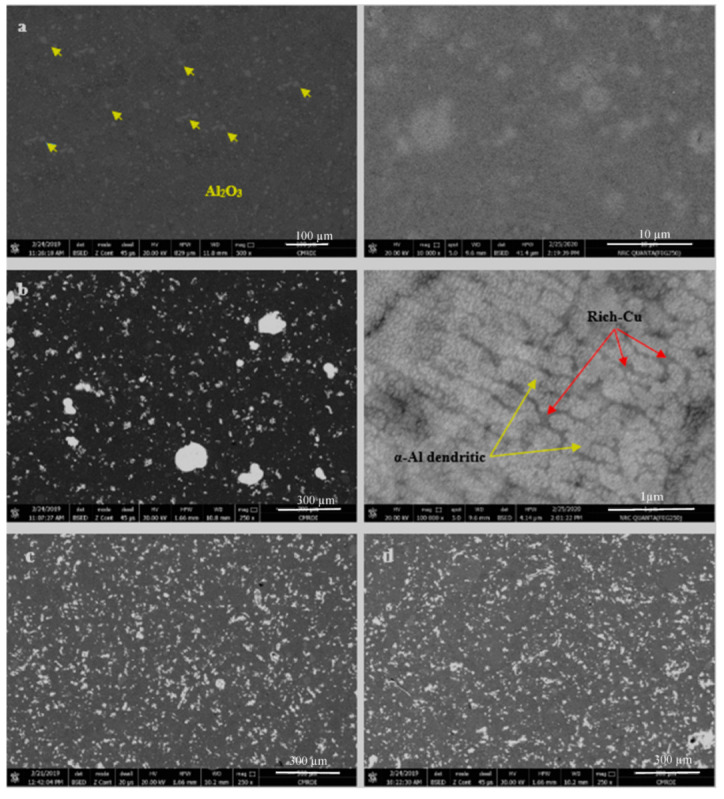
SEM (BSE) of hot-pressed materials. (**a**) Pure Al matrix, (**b**) Al-15 wt% Cu, (**c**) Al-15 wt% Cu/0.4 wt% GNPs-Ag, (**d**) Al-15 wt% Cu/0.6 wt% GNPs-Ag, (**e**) Al-15 wt% Cu/1.2 wt% GNPs-Ag GNPs, (**f**) Al-15 wt% Cu/1.8 wt% GNPs-Ag.

**Figure 6 materials-16-07174-f006:**
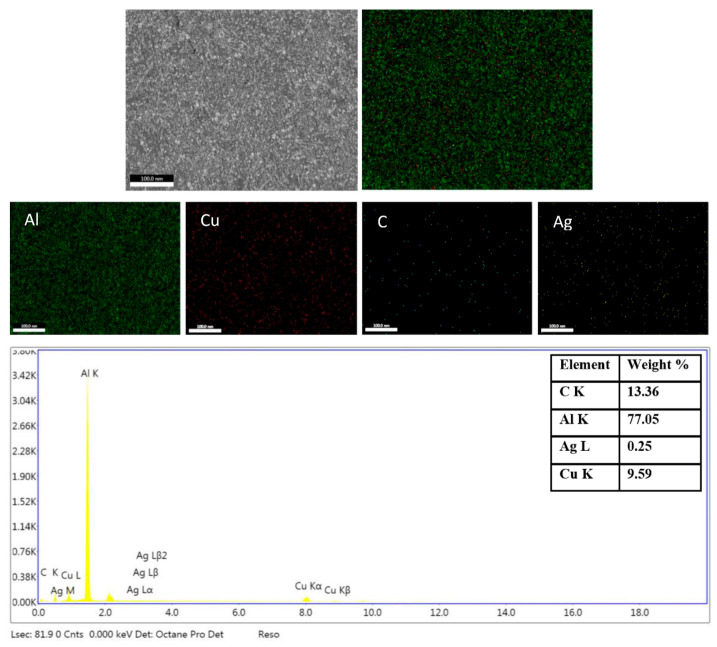
EDS mapping of Al-15 wt% Cu/1.8 wt% GNPs chemical composition.

**Figure 7 materials-16-07174-f007:**
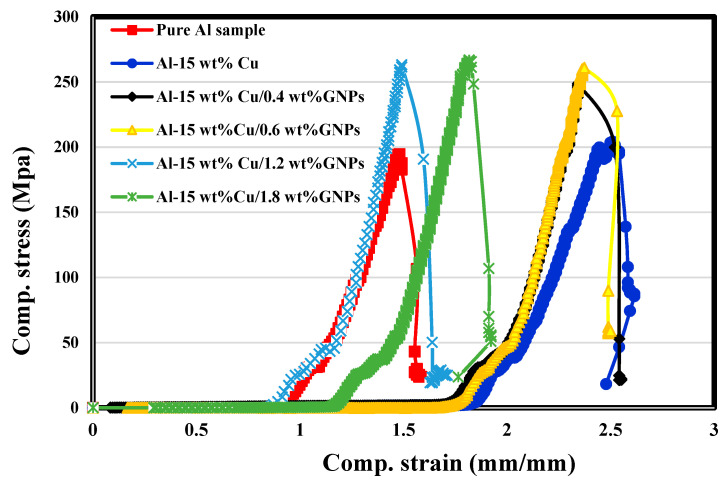
Compression stress–strain curves for pure Al, Al-15 wt% Cu, Al-15 wt% Cu/0.4 wt% GNPs, Al-15 wt% Cu/0.6 wt% GNPs, Al-15 wt% Cu/1.2 wt% GNPs, and Al-15 wt% Cu/1.8 wt% GNPs.

**Table 1 materials-16-07174-t001:** Composition of the nanocomposites.

Sample No.	Composition
1	Pure Al
2	85 wt% Al–15 wt% Cu
3	99.6 wt% (85% Al + 15% Cu)/0.4 wt% GNPs/Ag
4	99.4 wt% (85% Al + 15% Cu)/0.6% GNPs/Ag
5	98.8 wt% (85% Al + 15% Cu)/1.2% GNPs/Ag
6	98.2 wt% (85% Al + 15% Cu)/1.8% GNPs/Ag

**Table 2 materials-16-07174-t002:** Chemical composition of W320 steel.

C	Si	Mn	Cr	Mo	V
0.31	0.30	0.35	2.90	2.70	0.50

**Table 3 materials-16-07174-t003:** RD of the fabricated nanocomposites.

Material	Relative Density (%)
Pure Al	99.6%
Al/15% Cu	99.9%
Al-15% Cu/0.4 wt% GNPs	99.5%
Al-15% Cu/0.6 wt% GNPs	99.2%
Al-15% Cu/1.2 wt% GNPs	98.8%
Al-15% Cu/1.8 wt% GNPs	98.7%

**Table 4 materials-16-07174-t004:** HV Micro-Hardness of fabricated samples.

Material	Micro-Hardness (HV)
Pure Al	216.2
Al/15% Cu	230
Al-15% Cu/0.4 wt% GNPs	279.16
Al-15% Cu/0.6 wt% GNPs	286
Al-15% Cu/1.2 wt% GNPs	289.4
Al-15% Cu/1.8 wt% GNPs	328.42

**Table 5 materials-16-07174-t005:** Compression yield stress CYS (Mpa) and ultimate yield stress-fabricated samples UCS (Mpa).

Sample No.	CYS (MPa)	UCS (MPa)
Pure Al	35.113	194.43
Al + 15 wt% Cu	39.1	204.145
Al + 15 wt% Cu/0.4 wt% GNPs	41.685	248.6
Al + 15 wt% Cu/0.6 wt% GNPs	44.741	260.843
Al + 15 wt% Cu/1.2 wt% GNPs	45.734	263.21
Al + 15 wt% Cu/1.8 wt% GNPs	54.9932	266.9956

## Data Availability

The data presented in this study are available in this article.

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
