# Peer review of "Manufacturing of Aluminum Nano-Composites Reinforced with Nano-Copper and High Graphene Ratios Using Hot Pressing Technique"

_materials, 2023, doi:10.3390/ma16227174_

Round 1

Reviewer 1 Report

This manuscript reports the manufacturing of a duel matrix (Al-Cu) nano composite reinforced with high graphene ratios by hot pressing technique. Generally, the manuscript is well executed. The following minor adjustments and clarifications should be addressed before considering for publication:

 Page 2, second paragraph: too many details on the properties of graphene (which is not necessary). Might consider to make the description more precise.

 Page 3, line 138: A new paragraph could start from Despite these advantages…”

 Page 6, it would be better to present the fabricated sample before the section “3. Composites Characterization”.

Some grammatical errors:

Line 34: “The Ideal…”, please revise

 Line 231: “15wt%”, there should be a space between the number and the unit, please double check the other cases throughout the manuscript.

 Line 438, “reach to 305 MPa”, please revise.

Author Response

Response to Editor and Reviewer

Authors want to thank editors and reviewers for their careful and thorough reading of this manuscript and for the thoughtful comments and constructive suggestions, which help to improve the quality of this manuscript.

Comments from editors and reviewers

Reviewer 1

  1. Page 2, second paragraph: too many details on the properties of graphene (which is not necessary). Might consider to make the description more precise.

Reply:

The paragraph has been modified.

  1. Page 3, line 138: A new paragraph could start from “Despite these advantages…”

Reply:

The paragraph has been modified.

  1. Page 6, it would be better to present the fabricated sample before the section “3. Composites Characterization”.

Reply:

The fabricated samples image has been added.

Some grammatical errors:

  1. Line 34: “The Ideal…”, please revise

Reply:

Correction has been done.

The Ideal manufacturing composites combine the stiffness of the matrix and the reinforcement strength that have specific properties which don't belong to either alone.

  1. Line 231: “15wt%”, there should be a space between the number and the unit, please double check the other cases throughout the manuscript.

Reply:

Correction has been done.

  1. Line 438, “reach to 305 MPa”, please revise

Reply:

The Correction has been done.

Increasing the GNPs up to 1.8 wt%GNs ratio led to increasing the compressive strength to 266.9 MPa compared with 194.43 MPa and 204.1 for pure Al and Al/15Cu nanocomposites, respectively.

Thank you very much dear prof., for your interest and cooperation. 

Reviewer 2 Report

I appreciate the idea of this manuscript: to improve mechanical strength by adopting GNPs and Cu as dopants. However, the majority portion of this manuscript need to be re-written to make readers easier to follow. 

1. I think the heading of this manuscript needs to be re-written. To me, this manuscript focused on Al Nanoparticles, where Cu and GNPs are additive dopants. I think the authors should reflect this material system being studied in the heading. 

2. The introduction (Especially second and third paragraph need to be re-written).

       - Why need to introduce graphene? 

       - The third paragraph should include the experimental setup and background of this work, but the current version is hardly reflect this information. Besides, I suggest the author divided this paragraph into several shorter ones, the current version is very hard to follow.

3. Label is missing in Figure 7. 

4. Conclusion also need to be re-written, please avoid using bullet points. 

Author Response

Response to Editor and Reviewer

Authors want to thank editors and reviewers for their careful and thorough reading of this manuscript and for the thoughtful comments and constructive suggestions, which help to improve the quality of this manuscript.

Comments from editors and reviewers

Reviewer 2

  1. I think the heading of this manuscript needs to be re-written. To me, this manuscript focused on Al Nanoparticles, where Cu and GNPs are additive dopants. I think the authors should reflect this material system being studied in the heading.

Reply:

Heading this the manuscript has been re-written.

Manufacturing of Aluminum Nano-composites Reinforced with nano-copper and High Graphene Ratios using Hot pressing Technique

  1. The introduction (Especially second and third paragraph need to be re-written).

Reply:

Adding graphene, CNT, SiC, WC, or graphite in powder form to aluminum or copper metals improves their mechanical characteristics without affecting their physical properties. From an engineering point of view, the study of graphene and its derivatives has been focused on due to the novel properties of graphene and its potential use in multiple application fields. That is not limited to electronics, heat transfer, biosensing, membrane technology, battery technology, and advanced composites. Graphene has interesting features such as; large electric and thermal conductivity, high mechanical strength, and high optical transparency. These properties are only detected for graphene films formed of only one or a few layers [[7-9]. According to Changgu Lee, Graphene is the strongest material ever tested, with a Tensile strength of 130 GPa and a Young’s Modulus (defines stiffness) of 1 TPa [10]. Apart from this, Graphene is unbelievably light, weighing only 0.77 mg/m2 [11]. It has been experimentally proven that its electron mobility is almost independent of temperature [12]. Graphene produces a highly opaque atomic monolayer in a vacuum, as it can absorb approximately 2.3% of white light. Adding another layer of Graphene increases the amount of white light absorbed by approx. the same value (2.3%) [13].

A.Fathy et al. [14] studied aluminum reinforced with Al2O3-coated nickel and graphene nanoplatelets (GNPs) using the electroless deposition and hot pressing technique. The compressive strength of the composite containing 1.4% GNPs was 1.52 times greater than pure Al. Increasing the GNPs content reduced the wear rate of Al-Al2O3/GNPs nanocomposite to 19 times less for a sample containing 1.4% GNPs than for Al. A. Saboori et al. [15] studied the densification of Al matrix reinforced with various GNPs content (0, 0.5, 1.0, 1.5, and 2.0 wt%). The samples were sintered at diffrent temperatures (540, 580, and 620 oC) under nitrogen flow to determine the sinterability of the nanocomposites. It was found that by increasing the temperature, the sinterability of the nanocomposite increases due to easier diffusion. On the contrary, the sintering ability of the nanocomposite decreased with the increase of GNPs content up to 2 wt% due to the work-hardening effect caused by the GNPs. The Vickers hardness of sintered nanocomposites increased proportionally with GNPs content and sintering temperature. Duosheng Li et al. [16] successfully fabricated GNPs/Al nanocomposites containing 0.5 vol%, 1.0 vol%, and 2.0 vol% of GNPs. It was found that the lower content of GNPs in the composite induced better mechanical properties and better microstructural characteristics than pure aluminum. Adding 0.5 vol%, 1.0 vol%, and 2.0 vol% GNPs to the Al matrix gave the average compressive strengths of GNPs/Al nanocomposites 297, 345, and 527 MPa, respectively, which remarkably increased the strength over the original aluminum by 330% to 586%. Gang Li et al. [17] studied the graphene nanosheets (GNs) reinforced Al matrix composites fabricated by high-energy ball milling and vacuum hot pressing technique. The graphene content was (0.25 wt%, 0.5 wt%, and 1.0 wt%). A good interfacial bonding was obtained in GNs/Al composites. 38.27% and 56.19% increments in the yield strength and ultimate tensile strength of Al-0.25GNSs compared with pure Al were achieved. The elongation slightly decreased.

Furthermore, one of the common problems that limit the graphene application in Al (MMCs) is the poor wettability between them, which causes a low interface adhesion resulting in poor mechanical and thermal properties [19-20]. At the same time, the interaction between Al and C to form Al3C4 intermetallic phase reduces the mechanical properties [19, 21]. The electroless deposition method is one of the solutions that can be used to produce a wettability layer on the metal's surface to improve the interfacial bond between the metal and ceramic phases [22-25].

Hot Pressing is the setup in uniaxial hot pressing that is very similar to conventional PM pressing, except that heat is applied during compaction. The product is generally dense, intense, challenging, and dimensionally accurate [26-33]. Despite these advantages, the process presents some technical issues that limit its adoption. Principal among these is (1) selecting a suitable mold material that can withstand the high sintering temperatures; (2) a longer production cycle required to accomplish sintering; and (3) heating and maintaining atmospheric control in the process.

Because of the aforementioned knowledge, this work deals with the preparation of the aluminum-copper-graphene nanocomposites using electroless deposition and hot-pressing techniques. Scanning electron microscope (SEM) and X-ray diffraction (XRD) were used to study the formation of any new phases and the morphology of nanoparticles of the as-prepared composites. Density, hardness, and compressive strength were also estimated.

  1. Label is missing in Figure 7.

Reply:

The Label has been modified.

  1. Conclusion also need to be re-written, please avoid using bullet points. 

Reply:

The conclusion has been re-written.

  1. Due to the continued stirring of graphene during the cleaning and the coating with silver and copper and then mixing them with the nano-aluminum particles for 6 hr, a homogenous dispersion of it at high ratios was achieved.
  2. New intermetallic (Cu9Al4) between aluminum and copper was formed. The formed intermetallic enhances the grain boundaries' strength and, consequently, the fabricated aluminum matrix's hardness.
  3. Due to encapsulating the GNs/Ag layers with Cu, the interaction between them and aluminium was restricted, and no peaks for Al4C3 were detected.
  4. The high magnification of the microstructure showed that the GNP layers were transparent and in a horizontal position. Also, the adhesion at the edges of the GNP layers with the Al matrix was improved due to coating the GNPs with Ag, where the wettability was improved.
  5. The GNPs content significantly improved the hardness of the nanocomposite to reach 328.24 HV for samples containing 1.8% GNPs compared with 216.2 HV and 230 HV for the pure Al and Al/Cu samples, respectively.
  6. Increasing the GNPs up to 1.8 wt%GNs ratio led to increasing the compressive strength to 9 MPa compared with 194.43 MPa and 204.1 for pure Al and Al/15Cu nanocomposites, respectively.

Thank you very much dear prof., for your interest and cooperation. 

Reviewer 3 Report

-Please describe the unit of ratios in the abstract. volume or weight?

-For silver and copper elements, please use element symbols everywhere.

-Please give striking numeric results in the abstract section.

-'electroless coating' must added to Keywords.

-In the line 102, I think there is some wrong about the reference (fathy 2018) (30)

-The literature on the mechanism of electroless coating on the metal matrix composites could be expanded a little more. The following publications may be of interest to authors.

*Effect of Al2O3 content and milling time on the properties of silver coated Cu matrix composites fabricated by electroless plating and hot pressing. Materials Today Communications 24 (2020): 101153.

*Microstructure and wear characterization of Al2O3 reinforced silver coated copper matrix composites by electroless plating and hot pressing methods. Materials Today Communications 27 (2021): 102205.

-The novelty of the study should be clearly demonstrated in the last paragraph of the introduction section. What gap does this study fill in the literature?

-Between the line 171-174, the statements are interesting. For example, the sentence of 'Some GNPs sheets were rolled in the form of carbon nano-tube', why they were rolled?

- the sentence of 'Furthermore, coating with copper increases the separation of GNPs layers. It appears in image (d) transparent', how does the copper increase it and why is it transparent? Please give an explanation in the text.

-In Fig. 1e, the particles doesnt seem a nanoscale. Nanoscale term is for under 100 nm.

-In Fig. 2, there is a missing character in 'Surface Treatme..'

-How did you determine that the silver content in the structure is 5%wt.? Similarly, 15%wt. for copper?

-In the line 209, it is 'electroless' not 'non-electroplating'.

-Any process control agent was used in the milling process? If so, please describe.

-In the Figure 14 caption, the statements of Al-15Al seem wrong, it is not 15Cu?

 -The descriptions of Al,Cu, C.. given in the XRD graph are shifted and incomprehensible. Please revise it.

-hardness seems to increase greatly while relative density decreases. Can you explain this mechanism? In addition, it is very difficult to distinguish copper and silver particles in internal structure images, perhaps point EDS analysis can be given.

-Figure 7 must be revised, legends are shifted.

-Examination of the fracture surfaces (SEM and EDS) would have provided a better understanding of the mechanism by showing the electroless coating elements on this surface, which added originality to the study.

Author Response

Response to Editor and Reviewer

Authors want to thank editors and reviewers for their careful and thorough reading of this manuscript and for the thoughtful comments and constructive suggestions, which help to improve the quality of this manuscript.

Comments from editors and reviewers

Reviewer 3

  1. Please describe the unit of ratios in the abstract. Volume or weight?

Reply:

Corrections have been done.

  1. For silver and copper elements, please use element symbols everywhere.

Reply:

Corrections have been done.

  1. Please give striking numeric results in the abstract section.

Reply:

Numeric results have been added.

  1. 'electroless coating' must added to Keywords.

Reply:

The corrections has been done.

  1. In the line 102, I think there is some wrong about the reference (fathy 2018) (30)

Reply:

The correction has been done. The (fathy 2018) (30) has been omitted.

  1. The literature on the mechanism of electroless coating on the metal matrix composites could be expanded a little more. The following publications may be of interest to authors.

*Effect of Al2O3 content and milling time on the properties of silver coated Cu matrix composites fabricated by electroless plating and hot pressing. Materials Today Communications 24 (2020): 101153.

*Microstructure and wear characterization of Al2O3 reinforced silver coated copper matrix composites by electroless plating and hot pressing methods. Materials Today Communications 27 (2021): 102205.

Relay:

The references have been cited.

The density, hardness, tribological properties, and electrical conductivity of copper matrix composites reinforced with coated and uncoated Al2O3 were studied. The samples were fabricated using the hot-pressing process. In spite of decreasing the density of samples, the hardness was increased due to the hard structure of Al2O3 particles. The highest hardness was achieved for silver-coated copper matrix composites fabricated by using 2 h of milling at 3 wt%. Al2O3 ratio. The highest electrical conductivity (88 %IACS) was measured for the 3 wt% Al2O3 silver-coated copper sample synthesized by 2 h milling. The wear mechanism was abrasive wear with grooves and scratches. [26, 28].

  1. Between the line 171-174, the statements are interesting. For example, the sentence of 'Some GNPs sheets were rolled in the form of carbon nano-tube', why they were rolled?

Relay:

Some single GNPs sheets were rolled in the form of carbon nano-tubes during the coating process. This may be happened due to the large surface area of the graphene layer compared with its thickness. Furthermore, coating with copper increases the separation of GNPs layers. Before adding the reducing agent to the copper solution, the graphene layers were stirred for 5 min which encourages its exfoliation. It appears in the image (d) transparent. Deposited graphene layers with copper metal are evident from image (e). Copper was included in the form of a dendritic shape.

  1. The sentence of 'Furthermore, coating with copper increases the separation of GNPs layers. It appears in image (d) transparent', how does the copper increase it and why is it transparent? Please give an explanation in the text.

Relay:

The electroless coating process established homogeneous distribution of the Ag. Some single GNPs sheets were rolled in the form of carbon nano-tubes during the coating process. This may be happened due to the large surface area of the graphene layer compared with its thickness. Furthermore, coating with copper increases the separation of GNPs layers. Before adding the reducing agent to the copper solution, the graphene layers were stirred for 20 min which encourages its exfoliation. It appears in the image (d) transparent. Deposited graphene layers with copper metal are evident from image (e). Copper was included in the form of a dendritic shape.

  1. In Fig. 1e, the particles doesnt seem a nanoscale. Nanoscale term is for under 100 nm.
  2. Relay:

Scale for the nanoparticles has been added (<50 nm).

  1. In Fig. 2, there is a missing character in 'Surface Treatme..'

Relay:

The correction has been done.

  1. How did you determine that the silver content in the structure is 5%wt.? Similarly, 15%wt. for copper?

Relay:

This was done by weighing the powder before and after coating.

  1. In the line 209, it is 'electroless' not 'non-electroplating'.

Relay:

The correction has been done.

  1. Any process control agent was used in the milling process? If so, please describe.

Relay: (Experimental part)

The mixing process was achieved using a high-energy ball milling technique using alumina ceramic balls of 12 mm diameter in an acetone medium. 

  1. In the Figure 14 caption, the statements of Al-15Al seem wrong, it is not 15Cu?

Relay:

The figure caption has been modified.

  1. The descriptions of Al,Cu, C.. given in the XRD graph are shifted and incomprehensible. Please revise it.

Reply:

The correction has been done.

  1. Hardness seems to increase greatly while relative density decreases. Can you explain this mechanism? In addition, it is very difficult to distinguish copper and silver particles in internal structure images, perhaps point EDS analysis can be given.

Reply:

  • It is evident that by increasing the GNPs content, the hardness of Al/Cu nanocomposite is gradually increased up to 1.8 wt%, in spite of decreases in the relative density. As shown in Table 3, the reduction in the relative density by varying the GNs ratios was not high, which means that the reduction may be related to only the low density of the GNs layers. Porosity was absent as shown by the microstructure. The sample containing 1.8% recorded 328.24 HV compared to 216.2 HV and 230 HV for the pure Al and Al/15%Cu samples, respectively. This improvement is attributed to the GNPs layer's characteristics, which have extremely high strength and mechanical properties, also providing a high contact force for deforming during indentations. Another main factor that affects the hardness is the improvement of wettability between Al and GNPs by coating GNPs with Ag and Cu layers. In which proper distribution was achieved all over the Al matrix.
  • The copper and silver has been confirmed using the EDS mapping shown in Figure 6.
  1. Figure 7 must be revised, legends are shifted.

Reply:

The correction has been done.

Thank you very much dear prof., for your interest and cooperation.

Round 2

Reviewer 3 Report

Thanks the authors for their satisfactory revisions.